# Thiocoumarins: From the Synthesis to the Biological Applications

**DOI:** 10.3390/molecules27154901

**Published:** 2022-07-31

**Authors:** Maria J. Matos, Lourdes Santana, Eugenio Uriarte, Fernanda Borges

**Affiliations:** 1Departamento de Química Orgánica, Facultade de Farmacia, Universidade de Santiago de Compostela, 15782 Santiago de Compostela, Spain; lourdes.sanatana@usc.es (L.S.); eugenio.uriarte@usc.es (E.U.); 2Centro de Investigação em Química da Universidade do Porto (CIQUP), Departamento de Química e Bioquímica, Faculdade de Ciências, Universidade do Porto, Rua do Campo Alegre s/n, 4169–007 Porto, Portugal; fborges@fc.up.pt; 3Instituto de Ciencias Químicas Aplicadas, Universidad Autónoma de Chile, Santiago 7500912, Chile

**Keywords:** thiocoumarins, 2-thioxocoumarins, dithiocoumarins, synthesis, biological applications

## Abstract

Coumarin is a privilege scaffold in medicinal chemistry. Coumarin derivatives are still an emerging class of highly potent pharmaceutical drugs, best known in the field of antimicrobials and anticoagulants. Thiocoumarins are a particular class of coumarins in which one or two of the oxygen atoms are replaced by a sulfur. They are chemically subdivided in three groups: Thiocoumarins, 2-thioxocoumarins, and dithiocoumarins. This review emphasizes the rationale behind the synthesis and biological applications of the most relevant publications related to this family of compounds. Particular attention has been given to their potential as drug candidates, with particular emphasis in the last 5 years. This article is based on the most relevant information collected from multiple electronic databases, including SciFinder, Pubmed, Espacenet, and Mendeley.

## 1. Introduction

This review describes the preparation and biological activities of a particular family of coumarins, the thiocoumarins. The substitution of an oxygen atom by a sulfur is a common strategy in medicinal chemistry and drug discovery, to modulate the chemical properties of a molecule of interest. In fact, the electron deficient and bivalent sulfur atom presents two different areas of positive electrostatic potential due to the low-lying σ* orbitals of the C–S bond, available to interact with different electron donors as nitrogen or oxygen and, possibly, π-systems [1]. Recent studies discuss the potential of this phenomenon in drug design [1,2]. Therefore, several sulfur-containing functional groups are present in a broad range of drugs and natural products [3]. However, thiocoumarins have been receiving little attention from the research community compared with the coumarins, which are widely explored as scaffold in medicinal chemistry [4]. For example, searching in Pubmed for the keyword thiocoumarin, the last references are from 2022, and are only four, with one related to 4-sulfanylcoumarins. This may be due to impediments regarding their complex synthesis, scarcity of starting materials, bibliographical references, etc.

In the particular case of the coumarin scaffold, two different oxygen atoms can be replaced by sulfur, providing three different new scaffolds: Thiocoumarins, 2-thioxocoumarins, and dithiocoumarins (Figure 1).

In this review, special attention will be paid to the first group, which is most explored in the scientific bibliography. In addition, we will provide few highlights on the two other groups, to make this overview as complete as possible. Scientific articles and patents, collected from scientific sources, such as Pubmed, SciFinder, Espacenet, and Mendeley, are included in this review. Moreover, the most relevant synthetic pathways and biological applications are herein highlighted.

## 2. Discussion

### 2.1. Thiocoumarins

In the 1980s, thiocoumarins already had clinical use as haemorrhagic agents, anticoagulants (interfering with vitamin K-dependent coagulation factors), and antiallergic agents [5]. In fact, some thiocoumarin derivatives found important applications as anticoagulant rodenticides pesticides [6].

During the decade of the 1980s, the versatile synthesis from the acrylic and propiolic *ortho* esters and benzenethiols was described [7]. After those first reports, some other evidences on the interest of thiocoumarins as potential bioactive molecules have been published. However, a deeper understanding of biology and biological functions, and how these molecules really work, has to be acquired before considering thiocoumarins as a trend topic in medicinal chemistry. One of the most recent subjects of interest related to this scaffold is the role of thiocoumarins as carbonic anhydrase inhibitors [8,9,10,11,12]. The described thiocoumarins, in particular the 6-hydroxy-2-thioxocoumarin, proved to bound to the human carbonic anhydrase II active site in a completely different mode compared with coumarins, commonly hydrolyzed by the esterase carbonic anhydrase to the corresponding 2-hydroxycinnamic acids. The role of thiocoumarins inhibiting carcinogenesis has also been studied over the last decade [13]. Moreover, few indole derivatives were tested as potential drugs for PUVA photochemotherapy [14]. Furthermore, ethers of thiocoumarin, at position 4, have been reported as nitric oxide synthase inhibitors in the nanomolar range [15].

Synthetically, most of the described thiocoumarins are obtained from the 4-hydroxythiocoumarin. Recently, a review collected the relevant information on the utility of this scaffold in organic chemistry [16]. Detailed synthetic methodologies, structures, and chemical properties of the 4-hydroxythiocoumarin were mentioned in this overview, as well as its most interesting bioactivities. In fact, this compound is easily obtained and modified to achieve key educts for the synthesis of heterocyclic systems.

Few papers per year describe some new advances in the field, related to both synthesis and biological applications. In 2019, there are only four relevant references on this topic, with the first one being a chemical mechanistic approach on the formation of two-center three-electron bonds by the hydroxyl radical induced reaction of thioesculetin [17,18,19,20]. The second study is focused on the anticonvulsant activity of a new series of synthetic 4-amino-3-nitrothiocoumarins [18]. The traditional synthesis starting from the thiophenol and the malonic acid, in the presence of POCl_3_ and AlCl_3_ with prolonged heating, is commonly used to afford the 4-hydroxythicoumarin, in low yield (Figure 2) [21]. The yields described for the first time for this traditional reaction were between 50 and 90%. However, further reports for other substituted thiocoumarins were lower than these. In this recent manuscript [19], an alternative two-step method has been proposed by the authors (Figure 2). An intermediate dithiophenyl ester of malonic acid is formed in high yield, using moderate conditions. In a second step, this intermediate undergoes a cyclocondensation in the presence of AlCl_3_ as a Lewis acid to give the final product in moderate to high yield. This new method has been described to provide an overall yield of 60%, which is considerably higher than the previous reports. Therefore, this can be considered a step forward in the field.

The third paper published in 2019 is focused on the design and synthesis of new anticonvulsants [19]. 3-Nitrocoumarins proved to be more active than the studied thiocoumarins. Finally, the synthesis of 4-sulfonylthiocoumarins was also described in 2019 [20]. Starting from the 4-hydroxythiocoumarin, via DABCO-catalyzed direct sulfonylation of 1-sulfonyl-1,2,3-triazoles, new 4-sulfonylthiocoumarins were efficiently obtained. The main advantage of this method is the lack of transition metal catalysts and extra oxidants. 

As previously mentioned, most of the traditional synthetic routes to afford the thiocoumarin scaffold involve a Lewis acid, as AlCl_3_ (Figure 3A) [22]. This reaction may be followed by nickel-catalyzed intramolecular recombination fragment coupling of the thioester to afford the corresponding benzothiophene. To show the applicability of this methodology, the 5-methyl benzothiophene has been obtained via an intramolecular decarbonylative strategy. Previously, preparation of thiocoumarin has been easily achieved via Friedel–Crafts type intramolecular cyclization of the precursor [22].

Complex thiocoumarins have been synthetized from simple ones, and not only 4-hydroxythiocoumarin is used as starting material for the synthesis of more complex thiocoumarins. 4-Chlorothiocoumarin is also used for acylating thiocoumarins to afford γ-ketoenones (Figure 3B) [23]. The authors described a one-step organocatalytic synthesis of 4-acylcoumarins from 4-chlorocoumarin, reporting the first examples of nucleophilic substitutions at the β-carbons of enones to afford γ-ketoenones [24]. 

Few scientific papers described the use of simple thiocoumarins as starting materials for more complex structures. As an example, a novel, base-catalyzed and highly diastereoselective direct Michael addition-isomerization was described for the efficient synthesis of Rauhut–Currier-type adducts. An unexpected α-addition of γ-butyrolactam onto the 3-acylthiocoumarin derivatives was observed rather than the γ-addition, which is more common [24]. This reaction may end with obtaining chalcones. Despite being an interesting alternative, it has not been explored in further works. Another example is the synthesis of (±)-thia-calanolide A, successfully accomplished using 3,5-dimethoxythiophenol as starting material, following a six-step reaction in an overall yield of 4.5% [25]. The key reaction involved a Friedel–Crafts tigloylation of 5,7-dihydroxy-4-thiocoumarin. Microwave was presented as an alternative to include lateral chains at position 3 of the 4-hydroxythiocoumarin, via Mannich reaction, to achieve compounds with antibacterial properties [26]. This reaction is clean, versatile, and described with high yields. 4-Hydroxythiocoumarin is also the starting material for Michael reactions at position 3 of the scaffold (Figure 3C) [27]. A primary amine-derived organocatalyst modified with an ionic group for asymmetric Michael reactions of C-nucleophiles with α,β-unsaturated ketones was synthesized. In the presence of this catalyst and an acidic co-catalyst, 4-hydroxythiocoumarin reacted with benzylidene-acetones or cyclohexenones to afford the corresponding Michael adducts in high yields (up to 97%) and with reasonable enantioselectivity (up to 80%). 4-Hydroxythiocoumarin is also the starting material for an easy synthesis of 4-acetylthiocoumarin via very high α-regioselective Heck coupling on tosylates (Figure 3D) [28]. This has been described as an efficient and versatile reaction. 4-(4′-Aryloxybut-2′-ynylthio) thiocoumarins were obtained via tosylation of 4-hydroxythiocoumarin followed by mercapto-substitution and condensation with 1-aryloxy-4-chlorobut-2-ynes [29]. Via sequential thermal and catalyzed Claisen rearrangements, and starting with the 4-hydroxythiocoumarin, it is possible to achieve the thiocoumarin-annulated furopyran moiety [30]. Pyrrole derivatives, at positions 2,3 of the thiocoumarins, have been also described, following a tandem [2,3] and [3,3] sigmatropic rearrangement [31]. Thiophenyl derivatives have been described by the same group following a similar approach [32]. The synthesis of 3-aminomethylenethiocoumarins is another case of how to increase a lateral chain in thiocoumarins [33]. In the article, a three-component synthesis by condensation of amines, α-amino acids, ureas, and carbamates with 4-hydroxythiocoumarin, in the presence of tri-ethylorthoformate, is described. Even if there is a temptation to compare these reactions, most of them are specific for some substitution patterns, and may only give specific thiocoumarin derivatives. Therefore, it is interesting to have an arsenal of different options to be able to synthetize more derivatives to establish structure–activity relationships.

One of the biggest molecules achieved starting from simple thiocoumarins is the novel 6,6′-arylidene-bis-[5-hydroxy-9-methyl-2,3-diarylthieno[3,2-*g*]thiocoumarins]. As the biscoumarins, these molecules were tested for their antimicrobial profile [34]. Similarly, and inspired on a class of furocoumarins, the angelicins, thioangelicins were synthetized and studied for PUVA chemotherapy [35]. These are two cases in which coumarins inspired the study of similar families of thiocoumarins.

7-Hydroxy-4-methylthiocoumarin is the precursor for a pyrano [6,7] thiocoumarins, reported for their anti-HIV potential [36]. From this series, an excellent derivative, with an EC_50_ in the low nanomolar range, was described.

4-Mercaptothiocoumarin is also a versatile starting material. It can be alkylated with different propargylic and allylic halides under phase-transfer-catalyzed conditions in the presence of tetrabutylammonium bromide or benzyl triethylammonium chloride catalyst in dichloromethane/NaOH solution, at room temperature [37]. Then, these 4-thiopropynyl- and thioallylthiocoumarins can be the starting material for different derivatives.

A tandem sp3 C–H functionalization followed by decarboxylation of 2-alkylazaarenes with thiocoumarin-3-carboxylic acid has also been used to achieve 4-substituted 3,4-dihydrothiocoumarins [38]. This catalyst-free reaction afforded a series of 29 azaarene-substituted 3,4-dihydrothiocoumarins. Another catalyst-free tandem reaction, this time a Michael addition followed by decarboxylation, was also described to achieve 4-substituted thiocoumarins [39]. This reaction uses the same thiocoumarin-3-carboxylic acid as starting material, together with 2-methylindole, in the same conditions of 1,4-dioxane, at 120 °C, this time to give 3-indolyl-substituted 3,4-dihydrothiocoumarins.

Few thiocoumarins have also been explored in the design of novel carrier systems. The development of egg shell-like nanovesicles using the thiocoumarin-3-carboxylate has been described [40]. These drug delivery vehicles have been used as a potential carrier for the bacteriostatic antibiotic sulfamethoxazole. The spectroscopic studies as well as the growth inhibition of *E. coli* exhibit that this formulation leads to pH responsive sustained release of the drug. 

Thiocoumarins have been recently prepared using microwave irradiation, in a versatile Lewis acid-catalyzed reaction [41]. Functional groups as pyrimidine and 1,3,4-oxadiazole have been introduced in the scaffold. The technique has been described as efficient, selective, fast, and clean. The series of compounds has been reported for the anti-microbial and antioxidant activities. 

Thiocoumarins can also be converted to dithiocoumarins using Lawesson’s reagent, in a reaction with ~30–40% yield (Figure 4) [9,11,42]. This reaction may contribute to deeply exploring these analogues, which are less studied due to the scarcity and rarity of starting materials.

As some simple coumarins, thiocoumarins have been recently described as fluorescent probes and sensors. A thiocoumarin, named 7-(*N*,*N*-diethylamino)-4-trifluoromethyl-2-thiocoumarin, has been used as sensor for heavy metal pollution and bacterial contamination. Its specific thiocarbonyl scaffold as donor-π-accepter electronic properties, provides intriguing optical properties and several applicable functions [43]. A thiocoumarin-containing ratiometric fluorescent probe has been described for the simultaneous detection of hypochlorite and singlet oxygen [44]. Once again, the thiocarbonyl moiety has been highlighted as an excellent alternative to traditional probes. Moreover, these applications have been described for live-cell imaging, with the thiocoumarin fluorescent probe turned-on once inside the cells [45]. The potential of these molecules for imaging has been recently claimed in a patent [46]. These new applications may revolutionize the field, making thiocoumarins more visible to the academic community.

### 2.2. 2-Thioxocoumarins

As previously described for the thiocoumarins, the 2-thioxocoumarins have been studied as carbonic anhydrase inhibitors [10,47]. The authors presented the X-ray crystal structure of the 6-hydroxy-2-thioxocoumarin on the human carbonic anhydrase II active site, and these results revealed an unprecedented and unexpected inhibition mechanism for these new inhibitors when compared with isostructural coumarins. It was proved that the exo-sulfur atom can link to a zinc-coordinated water molecule, while other parts of the scaffold can establish important interactions with different amino acids of the binding pocket. Compared with simple coumarins with the same substitution patterns, the inhibition mechanism is totally different. These molecules proved to be hydrolyzed, occluding the entrance of the binding pocket. This is a step forward in the design of efficient carbonic anhydrase inhibitors.

7-Diethylamino-4-hydroxymethyl-thiocoumarin (thio-DEACM) caged molecules are recently attracting attention. The thio-DEACM as a caging group has great properties, such as rapid blue-cyan light responsiveness and avoiding UV irradiation of cells in combination with the absence of toxicity of the released photocage [48].

As several other coumarin derivatives, 4-methylthiocoumarins have been described as a multitarget-directed ligand for the treatment of Alzheimer’s disease [49]. A novel 4-methylthiocoumarin derivative has been studied against acetylcholinesterase, butyrylcholinesterase, BACE1, β-amyloid aggregation, and oxidative stress involved in the pathogenesis of this neurodegenerative disease.

### 2.3. Dithiocoumarins

Dithiocoumarins were more explored more than the above mentioned 2-thioxocoumarins. Few reports on new synthetic routes to achieve this scaffold have been recently reported [50]. Thiocoumarins, presenting a variety of functional groups, were prepared from the corresponding thiophenols and diketene via cyclocondensation [42]. Thionation of the obtained products with Lawesson’s or Davy’s reagent led to the corresponding thiono- or dithiocoumarins.

These compounds are also used as starting materials to obtain more complex molecules. For most of the reactions involving dithiocoumarins as starting materials, the 4-hydroxy derivative is the selected molecule. This molecule is being used since the 1980s, when an efficient protocol was described to prepare it [51]. Following this protocol, 2’-chloroacetophenones react with carbon disulfide in the presence of sodium hydride to form 4-hydroxydithiocoumarin anions (Figure 5). Kinetic protonation provides the desired 4-hydroxydithiocoumarins. Alkylation of these precursors may provide S-alkyl derivatives.

As previously mentioned, reactions using different dithiocoumarins as starting materials have been described. An oxidative cross-coupling reaction of 4-hydroxydithiocoumarin and amines/thiols using a combination of iodine and tert-butyl hydroperoxide (TBHP) to provide access to lead molecules for biomedical applications was recently reported (Figure 5) [52]. This simple and versatile reaction allowed the attainment of a big family of compounds: Sulfonamides, disulfides, and sulfides. The authors claim an unprecedented, fast, mild, and environmentally friendly S–C bond formation, in addition to S–N and S–S bonds, in moderate to excellent yields.

A reaction starting from different dithiocoumarins, in the presence of nitroalkenes allowed for the formation of tricyclic molecules [53]. This reaction is catalyzed by potassium carbonate, which allows the closure of the new five members ring. This is an unprecedented and efficient method via a thio[3+2] cyclization reaction of 4-hydroxydithiocoumarins and trans-β-nitrostyrenes. This protocol has been described as faster than the previously reported, following mild conditions, presenting good yields, allowing for the formation of C–C and C–S bonds in a regioselective manner (Figure 6).

As for thiocoumarins, 4-hydroxy-dithiocoumarins are the most studied starting materials for obtaining new and more complex molecules. The reactions and yields are similar to the previously described analogues.

Few other authors described synthetic routes using different dithiocoumarins as starting materials to obtain more complex molecules (usually presenting extra rings). Few of these reactions follow green chemistry, claimed as more efficient, faster, and regio- and stereoselective [54,55]. Knoevenagel-hetero-Diels–Alder reaction as catalyst-free reaction was also described [56]. Novel pentacyclic thiochromone-annulated thiopyranocoumarins were obtained starting from 4-hydroxydithiocoumarin and *O*-acrylated salicylaldehydes. The main advantage is that this reaction is high regio- and stereoselective.

Regarding the application of these compounds, the use of optically active devices has been described and patented [57,58]. 3-Aryl derivatives and hydrophilic compounds are particularly interesting for this application. 

As described for thiocoumarins and 2-thioxocoumarins, dithiocoumarins have been studied as carbonic anhydrase inhibitors [11]. In fact, the reported review describes the synthetic methodologies and application of coumarins and their bioisosteres as carbonic anhydrases inhibitors. The most important chemical features to increase the activity of these molecules is described. A patent also claims the use of these molecules with this same activity and their application in the treatment of cancer [59].

Finally, the use of dithiocoumarins as herbicidal agents, has been described and patented [60,61]. 4-Phenoxy derivatives have been closely related to this activity. They were obtained starting from the well-known and used 4-hydroxydithiocoumarins.

## 3. Conclusions

Even if sulfur-containing functional groups are present in a broad range of drugs and natural products, thiocoumarins have been rarely explored as drug candidates. Few examples of synthetic routes, as well as the lack of starting materials, may be responsible for the lack of biological studies related to this family of compounds. Furthermore, coumarins have been extensively explored in different scientific fields, overshadowing their sulfur analogues. The most relevant examples of described thiocoumarins are compiled in this review. The role of thiocoumarins as chemical probes and sensors may provide a new life to these compounds for imaging and diagnosis purposes. 

## Figures and Tables

**Figure 1 molecules-27-04901-f001:**
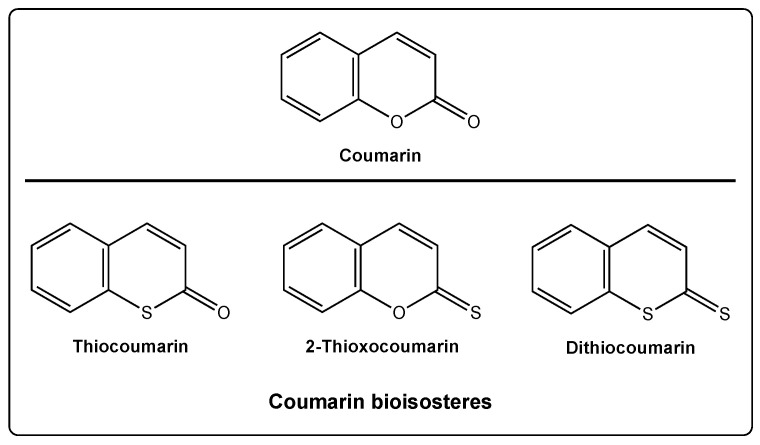
Coumarin and its bioisosteres: Thiocoumarin, 2-thioxocoumarin, and dithiocoumarin.

**Figure 2 molecules-27-04901-f002:**
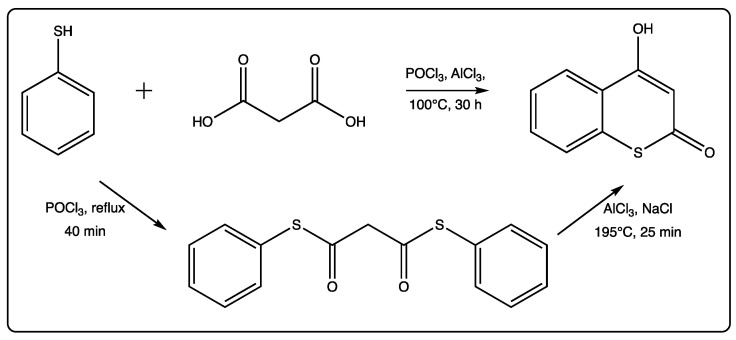
Alternative method of cyclocondensation of thiocoumarins.

**Figure 3 molecules-27-04901-f003:**
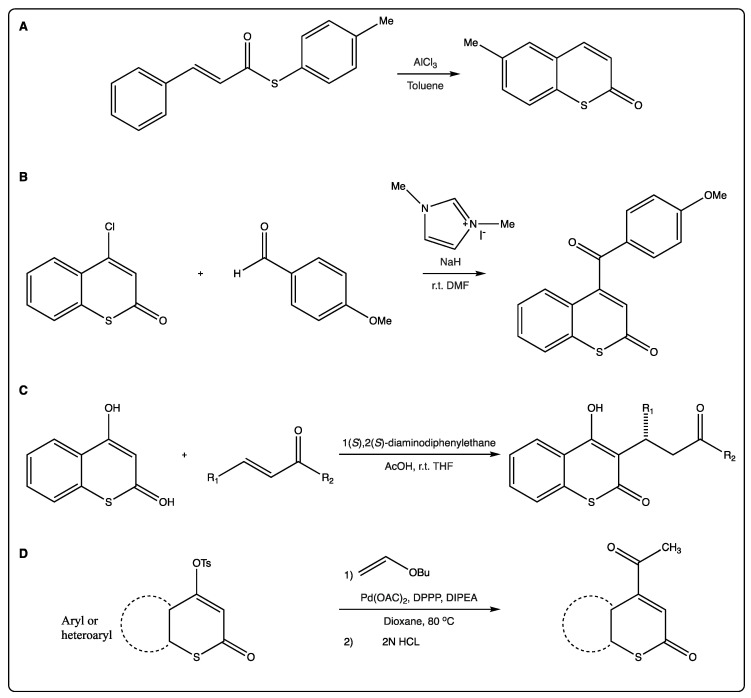
(**A**) Synthesis of thiocoumarins via traditional Lewis acid formation. (**B**) A straightforward organocatalytic synthesis of 4-aroylcoumarins. (**C**) Michael reactions at position 3 of the scaffold. (**D**) α-Regioselective Heck coupling of tosylates.

**Figure 4 molecules-27-04901-f004:**
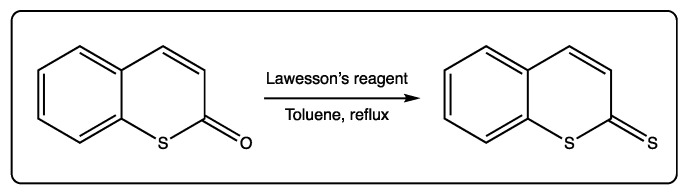
Conversion of thiocoumarins to dithiocoumarins.

**Figure 5 molecules-27-04901-f005:**
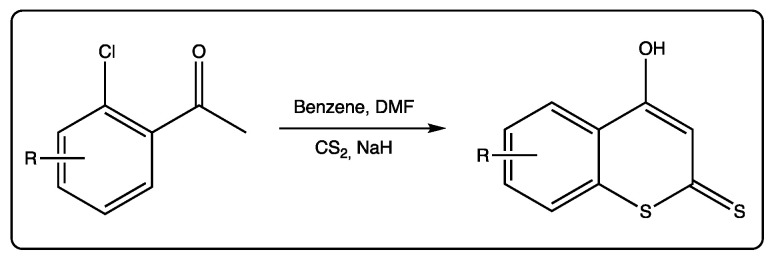
Synthesis of 4-hydroxydithiocoumarins starting from 2’-chloroacetophenones.

**Figure 6 molecules-27-04901-f006:**
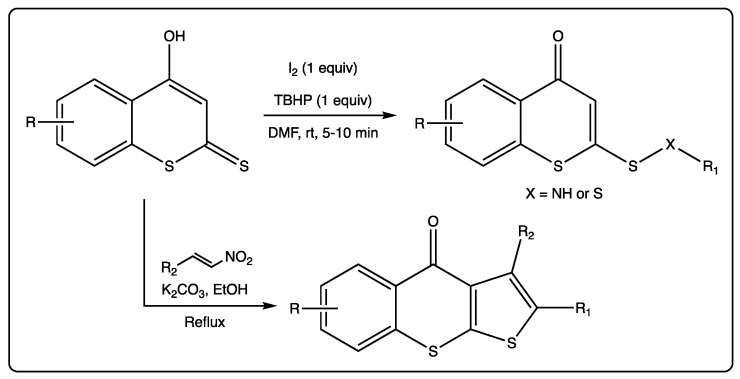
Thio[3+2] cyclization reaction of 4-hydroxydithiocoumarins and trans-β-nitrostyrenes.

## Data Availability

Not applicable.

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
