# Peer review of "Thiocoumarins: From the Synthesis to the Biological Applications"

_molecules, 2022, doi:10.3390/molecules27154901_

Round 1
Reviewer 1 Report
In the manuscript, Matos and co-worker described Thiocoumarins: From the synthesis to the biological applications. It’s not well written, no scheme is drawn and even references are also written wrong for e.g. Page2, line 81, the reference shouldn’t be 11.
Based on my judgement, the quality of this manuscript, at current stage, could not meet the requirements to publish in Molecules Journal.
1) The manuscript is not well written, they need to rewrite the manuscript and to draw chemical reactions for the schemes of the syntheses. Then this manuscript will be beneficial and easier to read for readers.
2) To make this manuscript more useful for readers they need to explain more briefly about biological activities as the title of the manuscript includes biological applications.
3) They also need to write down novelties and limitations for each synthesis.
4) Page 2, line 81, reference 11 should be replaced with reference 19
5) Page 3, line 92, reference 12 should be replaced with reference 20 6) Page 3, line 93, reference 13 should be replaced with reference 21
Author Response
We thank Reviewer 1 for the detailed evaluation of our manuscript. All the comments have been considered to improve the new version of the manuscript. We really appreciated all the efforts to deeply revise this review.
Answer 1: Examples of the chemical reactions are included in the manuscript. We added some more details, and modified some figures to make them more complete and clearer for the readers. Figure 3 has been improved, including some examples of reactions starting from simple thiocoumarins.
Answer 2: The most relevant biological applications of thiocoumarins over the last years are included in the manuscript. Some more details have been included.
Answer 3: Some more details on the synthetic methodologies have been included. The methodologies have been compared, as suggested.
Answer 4: Thank you for pointing out this detail. The mistake has been corrected, as suggested.
Answer 5: Thank you for pointing out this detail. The mistake has been corrected, as suggested.
Reviewer 2 Report
The present review summarizes the methods for the synthesis of sulfur isologues of coumarins. As an introductory remark, their classification, their availability, and some of their properties are described.
The review then focused on the synthetic methods for thioacoumarins. The review also touched the syntheses of thioxocoumarins and dithiocoumarins. But unfortunately, the review does not involve the actual chemical equations for the synthesis, and it is almost impossible for readers to draw those.
In addition, as one of important roles of the review describing the synthetic methods, the following should be referred, i.e., the scope and limitation of each reaction, and which methods are more readily available.
As a result, the review is not informative at all, and requires major revision.
Author Response
We thank the reviewer for the detailed evaluation of our manuscript. All the comments have been considered to improve the new version of the manuscript. We really appreciated all the efforts to deeply revise this review.
Answer 1: More information on the most relevant synthetic pathways has been added.
Answer 2: Some more details on the different methodologies have been included. Figure 3 has been modified in order to include some reactions using thiocoumarins as starting materials for more complex molecules.
Round 2
Reviewer 1 Report
Thanks Matos and coworkers for revising the manuscript. Appropriate changes were made.
I would recommend to publish in Molecules Journal.